# Assessment Tests in the Mathematics Teaching Guides in Spain. Analysis of the Content Blocks and the Treatment of Arithmetic Word Problems

**Raúl Tárraga-Mínguez \*** , **Julio Tarín-Ibáñez** and **Irene Lacruz-Pérez**

Department of Education and School Management, Faculty of Teacher Training, University of Valencia, 46022 Valencia, Spain; Julio.Tarin@uv.es (J.T.-I.); Irene.Lacruz@uv.es (I.L.-P.)
\* Correspondence: Raul.Tarraga@uv.es

**Abstract:** The teaching guides that complement textbooks have key importance in the assessment of competence in problem solving, because these materials contain the assessment tools that teachers frequently use to quantify the achievements of their students. In this paper, we set two aims: to analyze which curriculum contents are given priority in the assessment tests of the teaching guides; and to check to what extent these tests assess the steps of the mathematical problem solving process. For this, an analysis of the initial and final assessment tests of six Spanish publishers was conducted. The results show that the distribution of mathematical tasks by type of content does not fully conform to the theoretical framework proposed by TIMSS. In addition, only one of the six publishers considered the problem-solving process as evaluable.

**Keywords:** curriculum content; teaching guides; textbooks; verbal arithmetic problem solving

## 1. Introduction

From the pioneering studies carried out by the sociologist and pedagogue MW Apple in the 1980s to the present, research has highlighted that textbooks are usually the quintessential teaching material in most educational systems in advanced countries [1–3]. In this way, publishers play an important role in the process of interpreting, redefining, and selecting the official curriculum, becoming one important mediator when defining the real curriculum at school [4]. For this reason, authors such as [5] (p. 49), when he talks about the different curricular concretions, refers to textbooks as "the curriculum created to be consumed by teachers and students", so that textbooks constitute "authentic translators of the curriculum as a project and as a text embodied in specific practices" [6] (p. 33).

Thus, different studies confirm that textbooks are one of the materials most frequently used by teachers. Specifically, in the field of mathematics, the textbook occupies a solid position in the teaching–learning process of this area of knowledge [7,8]. Moreover, research indicates that this artifact is the main curricular resource used by teachers in daily classroom practice. In fact, mathematics textbooks determine largely what teachers teach and, consequently, an important part of what students learn, because, in many cases, their role is even more decisive than the prescriptions of the official curriculum [9–11].

In Spain, textbooks have always been one of the most widely used educational materials in school, even the only one sometimes [12]. According to the National Association of Book and Teaching Material Publishers [13], 71% of families consider that textbooks are essential in the education of their children, and 81% of teachers admit that they use them quite a lot or a lot in their daily work. In addition, since 2006, when the Organic Law of Education [14] came into force in our country, the publishing of textbooks has not required supervision and authorization by educational administrations. In this way, publishers become one of the most decisive agents in determining the real curriculum, which is established on the basis of the pedagogical beliefs of an author or a group of authors.

The reasons that lead teachers to use textbooks are diverse and complex. Thus, [15] explains that the deficient initial and continuing training of teachers does not enable them to face their own teacher practice autonomously; hence, they need to resort to "pre-elaborated curriculum materials" that facilitate their job. In this sense, [16] (p. 63) claims that for teachers, "textbooks act as a navigation map that reduce the uncertainty and complexity of teaching". According to other authors such as [17], there is a technocratic conception of the teaching profession that predominates in school, whose greatest exponent is the textbook. From this rationale, homogeneous curricular designs that are valid for any context and student are advocated; they must be thought out and elaborated by experts, and teachers must apply them routinely and mechanically. Finally, the Spanish educational system's policies of free textbooks naturalize the presence of this resource in schools as something typical of pedagogical normality, making it the quintessential material. In this regard, the authors of [18] (p. 266) ask, "Why are textbooks free, and other materials and resources are not?"

However, the textbook does not appear in the classroom as an isolated element. The didactic guide or teacher's guide that complements it also constitutes a fundamental tool when establishing the curriculum truly taught and assessed in educational practice. Indeed, in Spain, after the enactment of the General Education Law in 1970 [19], a new publishing modality came into play on the educational scene: the didactic guides. From that moment a new stage began, and it is still in force, in which the didactic guide, and not the teacher, is in charge of designing and developing the prescriptions of the official curriculum: what, how, and when to teach and assess. Therefore, we cannot consider textbooks and teaching guides as independent curricular materials. Currently, the publishing industry directs its political economy to elaborate "macro-projects" that include a multitude of materials and resources presented in different formats. Hence, the importance of expanding the field of study and directing the focus of attention toward the analysis of the didactic guides, as they are a key element that allows us to study the curriculum in practice.

Even though there are many national and international studies that have analyzed the role of problem solving (PS, henceforth) in textbooks [20–22], to our knowledge, these analyses have not been carried out in the didactic guides, a type of resource that has a clear impact on the day-to-day life of the classroom and thus deserves additional study [23].

In this paper, we address the analysis of the assessment tests presented in the didactic guides of the Spanish mathematics textbooks in two specific aspects: (1) the distribution of the items by type of curriculum contents that the assessment tests include, and (2) the treatment given to the assessment of the problem-solving process.

## 1.1. Curricular Frameworks for the Study of the Contents of the Mathematics Area

In Spain, the Royal Decree 126/2014 of February 28, which establishes the basic curriculum of primary education [24], includes five content blocks in the area of mathematics: processes, methods, and attitudes in mathematics; numbers; measure; geometry; and statistics and probability.

According to [24], the block of processes, methods, and attitudes in mathematics constitutes the cornerstone of the rest of the content blocks. For this reason, this block has been formulated with the aim of being part of the daily classroom tasks in order to work on the rest of the contents. The different steps that constitute the process of solving a problem, which are articulated around a method or model, as well as the development of a positive attitude toward mathematics, are part of this block.

The block of numbers include two conceptual categories: development of number sense or number literacy, and operability. The first category includes reading, writing, and ordering of different types of numbers (natural, fractions, decimals, roman). The second category includes knowledge, use, and automation of the algorithms of addition, subtraction, multiplication, and division with different types of numbers; calculation, taking into account the hierarchy of operations and applying their properties; and initiation in the use of percentages and direct proportionality.

In the measurement block, two conceptual categories are also contemplated: first, the knowledge, the use, and the transformation of the different measurement units related to length, surface area, weight/mass, capacity, time, and monetary systems; second, the use of the most relevant measuring instrument according to the type of measurement.

The block of geometry is organized into a single conceptual category focused on the knowledge, use, classification, reproduction, and representation of objects on the plane and in space. The basic geometric notions are part of this block; flat figures and the calculation of their areas; and polyhedrons, prisms, pyramids, and round figures.

Finally, the statistics and probability block is structured around two conceptual categories. One category comprises contents that allow the processing of information: the collection and recording of quantifiable information; the use of graphic representation resources such as data tables, bar blocks, and line diagrams; and reading and interpreting graphical representations of a data set. As its second category, this block also includes the contents related to the prediction of results and the calculation of probabilities.

At the international level, two frames of reference use the curriculum and its organization as a key element that contributes to the analysis of the level of knowledge and cognitive skills acquired by students in the area of mathematics. We refer to the Program for International Student Assessment (the PISA program) [25] in secondary education and Trends in International Mathematics and Science Study (TIMSS) [26] in primary and secondary education. Both stand out for being the most consolidated and having the greatest international follow-up. Although both have a very similar structure, we have focused on the theoretical framework proposed by TIMSS [26] due to the purposes of our work, because our subject of study is focused on the area of mathematics in primary education.

TIMSS [26] aims to assess the level of achievement of students, compare the results between different participating countries, and explain the differences detected according to the different educational systems. These objectives constitute what in this framework is called the attained curriculum. However, to achieve these objectives, TIMSS [26] starts from a first curricular level (official or intended curriculum) that is compared to a second curricular level (implemented curriculum).

Precisely, one of the aims of this study is to compare the official curriculum of the Spanish educational system with the implemented curriculum by textbooks; hence, the theoretical framework offered by TIMSS [26] constitutes a relevant reference for our research.

In Spain, Royal Decree 126/2014 [24] does not prescribe to what extent each content block must be present in the curriculum (beyond the mention that the first block must be the cornerstone of the rest of the blocks). However, the TIMSS [26] proposes a gradation regarding the distribution of individual blocks of the curriculum, which are called dimensions of knowledge, referring to the mathematical content involved in the task.

Specifically, in the test for the 4th grade of primary education, TIMSS gives a weight of 50% of the total of the test to the dimension of numbers, the dimensions of measurement and geometry are given 15%, respectively, and the statistics block is given the remaining 20% [26].

### 1.2. Problem Solving Models

Solving a problem is a cognitively complex task in which a set of skills, strategies, steps, or stages come into play. When explaining the process that the solver carries out during the PS, two types of theoretical models have been developed: on the one hand, general models based on heuristics; on the other hand, models from the field of cognitive psychology.

Models of the first type try to explain through a series of general or heuristic steps how students solve any type of problem (arithmetic, algebraic, geometric, etc.). Therefore, their main advantages are breadth and flexibility, as they can be adapted to different problem situations.

It is considered that the model based on heuristics proposed by [27] was the pioneer in the description of the phases of solving problems; however, [28] had already described in 1933 the stages of thought in the process of PS: (1) Identification of the problematic situation; (2) precise definition of the problem; (3) means-ends analysis and solving plan;

(4) execution of the plan; (5) assumption of consequences; and (6) valuation of the solution, supervision, generalization.

According to [29], the stages proposed by [28] are a "prelude" or precedent to those proposed by [27] in 1945, "an action guide" so that the teacher could help the students in the PS process effectively: (1) Understand the problem; (2) design a plan; (3) execute the plan; and (4) verify the solution obtained. In any case, Pólya's model [27] was a reference for the later development of heuristic models in which different phases are proposed to solve problems: a first phase of analysis and understanding of the information provided in the problem statement; a second planning phase aimed at finding the right strategy, that is, an action plan; a third phase that is more automatic and closely linked to the previous one, in which the plan is executed by applying the corresponding algorithm; and a fourth, final phase of verification-reflection of the process followed and the meaning of the result obtained.

Another model based on heuristics was the one proposed by [30,31], who established four phases, just as [27], to (1) analyze and understand the problem; (2) design and plan the solution; (3) explore solutions; and (4) verify the solution. The main novelty of this model in comparison with the previous one was the consideration of other dimensions, apart from the heuristics, that are necessary to solve a problem, such as the prior knowledge of the solver, a control that allows the efficient use of available resources, and the belief system that students have about mathematics.

As we stated previously, the advantage of models based on heuristics is that they can be applied to any type of mathematical problem. However, their main limitation is that they make very generic descriptions of the steps that form the PS process. Thus, for example, in the first phase of the model proposed by [27] (to understand the problem), which type of understanding is necessary? Is it necessary to have a conceptual or mathematical comprehension, or a textual comprehension? It must be remembered that verbal arithmetic problems of additive structure, which we will refer to in this article as "word problems", have double character: mathematical and textual. In this sense, a mathematical equation with numerical data that must be solved by applying one or more mathematical operations underlies every problem, hence the mathematical nature of the problem. For instance, "Alberto had 54 euros. He got 57 euros more. He has spent a few euros and in the end, he has 13 euros left over. How much money has Alberto spent?" ($54 + 57 - X = 13$). That said, if the solving of the problem ends up with the execution of one or more algorithms that result in a numerical answer, the first step should be a comprehensive reading of its statement. Hence, different authors consider word problems as short texts, verbal or textual descriptions, authentic discursive entities, specific textual genres or textual units [32–37]. In short, against the vagueness (generality) of heuristic models, cognitive models, which are much more specific, do not limit themselves to general descriptions of the PS process, but rather delve into the mental processes involved in the task of solving a problem. Therefore, for these models, solving the problem in the previous example or any other word problem goes beyond mathematical knowledge. A deep comprehension of the problem involves understanding the problem as a mathematical structure and as a text. So much so that for the solving, "it is necessary to build a bridge: a link between the semantics of the language of mathematics and the semantics of natural language is required" [33] (p. 112). That is, if the students do not understand the problem as a textual statement, they will not be able to extract the underlying mathematical essence.

On another note, the different cognitive models also propose a set of stages or steps to solve problems. All of them agree on the idea that solving a problem is a complex task in which it is necessary to activate sophisticated strategies to understand the statement of the problem. This is the common thread of all cognitive models, even though these models differ from each other depending on the emphasis that each places on one or another aspect of the solving process. One of the most relevant models, which belongs to the latest generation of cognitive models, is the one proposed by [38]. In the model of these authors, the emphasis is on the importance of both quantitative (mathematical) and qualitative

(situational) understanding to solve word problems. Thus, solving the problem requires a numerical answer, but also a realistic answer that the solver acquires by reasoning about the context of the problem and based on the previous knowledge. Therefore, to solve the following problem: "A man needs a rope that is long enough to stretch between two poles that are separated 12 m from each other, but he only has 1.5 m lengths of rope. How many pieces will he need to bind in order to spread the rope between the two poles?" mathematical knowledge (division algorithm) is not enough, because it must be taken into account that, when knotting the rope between the posts, a few centimeters of length are lost. Therefore, it is necessary to apply a reasoning that goes beyond the strictly mathematical (situational context), because if we base the solving of this problem exclusively on an arithmetic procedure, it will lead us to an answer that, being correct from a mathematical point of view, will be meaningless.

Because not all problems involve the same level of difficulty, in the cognitive model of [38], two modes of solving are distinguished: the superficial mode and the genuine mode, whose main difference lies in the promotion, or not, of reasoning. Solving a problem using the superficial mode can be carried out in three steps: (1) selection of numerical data of the problem; (2) execution of the corresponding operation; and (3) expression of the result. For example, in the problem: "Andrés has 36 candies and Daniel has 24. How many less candies does Daniel have than Andrés?" even though it is possible to reason, the problem can be solved in three steps: (a) select the data from the statement without the need to understand it (36 and 24); (b) deduce the operation using a superficial strategy, specifically, the "keyword" strategy [39–41], also called the direct or literal translation strategy [42], which consists of selecting the data of the problem, "holding onto the numbers", according to the expression of [43], and operating with them taking as reference the linguistic terms of the statement (in the example problem: less than = subtract); and (c) report the result without checking if it is plausible from a mathematical or situational point of view. The superficial solving mode allows the solving of the easiest word problems, that is, the so-called consistent problems in the terminology of [44], in which there is a consistency or coherence between the superficial structure of the problem and the algorithm that must be applied. In this way, terms such as "win", "collect", "more", etc. imply a sum, while other terms such as "lose", "reduce", "less", etc. involve a subtraction. According to these authors, the following word problems are consistent: problems of combination 1; change 1, 2, and 4; comparison and equalization 2, 3, and 4.

Conversely, mathematically difficult or inconsistent problems require a schematic knowledge of the part–whole structure that organizes the relationships between quantities. For example, in the problem "Andrés has 29 candies and his brother Daniel has 44, how many candies does Daniel have more than Andrés?", the use of the "keyword" strategy would lead the solver to a wrong answer. Therefore, to solve this problem, the student needs more advanced mathematical knowledge; in this case, it is necessary to reason that, if Daniel has more candies than his brother Andrés, to know how many more he has (the difference), it is necessary to subtract, although in the surface structure of the problem statement appears the expression "more ... than". According to the dichotomous classification established by [44], inconsistent problems are those in which there is no coherence between the surface structure of the problem and the algorithm necessary for its solving: combination problems 2; change 3, 5, and 6; comparison and equalization 1, 5, and 6.

Moreover, a second factor that determines the difficulty of word problems is the location of the unknown; so that the problem will be more difficult, the further to the left the unknown is located. For instance, change problems have a maximum difficulty when the unknown is the initial set; the difficulty is less when the unknown appears in the change set; and even less when the unknown refers to the final set [45].

Finally, without diminishing the importance of the previous classifications, it should be clarified that the difficulty of a problem cannot be considered in absolute terms, because the perceived difficulty of a problem also depends frequently on the knowledge and experiences of individuals [46–48]. In this sense, [49] proposed in 1985 the concept of

threshold of problematicity, which will be different for each person. In this regard, [50] also stated that the existence of difficulties is not only an intrinsic characteristic of the problem, because the ease or difficulty in solving also depends on the prior knowledge of the solver. The author of [50] places the problem threshold according to the subject who faces the problem; if he masters all the necessary concepts and procedures, he will be in front of an exercise, while if he does not know them, he will have a problem.

In short, solving problems is a fundamental tool typical of advanced societies that all citizens must master in adulthood and that is mainly developed in school. Considering that the teaching guides that complement the textbooks contain the assessment tests of the students, an analysis of these curricular materials will allow us to know the publishers' treatment of PS. Therefore, given the relevance of these materials as documents that include the students' assessment tests already prepared, the main contribution of this study is to analyze the treatment of the word problem solving process in the assessment tests of the mathematics area of primary education, published in the teaching guides of six of the main publishing houses in Spain. Specifically, two objectives are proposed:

1. To analyze which content blocks are given priority in the assessment tests of the mathematics didactic guides.
2. To test to what extent these tests assess the different steps of the PS process, including reasoning as a key step, and what type of solving modes they propose to assess this process (superficial mode or genuine mode).

## 2. Materials and Methods

### 2.1. Sample

The study was carried out with the mathematics teaching guides of six Spanish publishers (Santillana, Anaya, S.M, Vicens Vives, Edebé, and Edelvives) published between the years 2014–2015 when a modification of the national education law came into effect [51]. The analysis focused on the assessment tests that each publisher proposes to assess the mathematical learning of the students at the beginning and at the end of the six school years of primary education (that is, two tests each year). In addition, the Santillana publisher includes for each grade a complementary final assessment test for an advanced level. Therefore, the final number of tests analyzed was 78. The total of items included in these assessment tests was 1616. Likewise, the different items of the set of assessment tests were analyzed globally, without specifying the educational level of each, as our aim was to study the treatment of the contents and mathematical processes in the whole stage of primary education.

### 2.2. Procedure

First, the different items of the assessment tests were classified taking as a reference the assessable learning standards of the five content blocks of the mathematics area, published in Royal Decree 126/2014 [24]. The learning standards arise from the assessment criteria, and these in turn from the contents established by the official curriculum [51]:

(a) Block 1. Processes, methods and attitudes in mathematics (e.g., "On the volleyball team there are 5 girls with brown hair, 3 blonde girls and 1 redhead girl. How many girls are there on the volleyball team? Draw, order the steps you have to follow to solve the problem and solve it. Then, explain orally how you did it" (Edelvives. Didactic Guide, 1st grade).

(b) Block 2. Numbers (e.g., "Multiply in two different ways and compare the results (Data: 6, 8 and 9). What property have you applied?" (Anaya. Didactic Guide, 4th grade).

(c) Block 3. Measurement (e.g., "Indicate which of the following numbers correspond to 2 h 6 min: (a) 136 min; (b) 7560 s; (c) 120 min 360 s; (d) 126 min" (H.M. Didactic Guide, 5th grade).

(d) Block 4. Geometry (e.g., "Identify every regular polyhedron. Indicate how many sides it has: tetrahedron, dodecahedron, and icosahedron" (Santillana. Didactic Guide, 6th grade).

(e) Block 5. Statistics and probability (e.g., "You flip two coins. Write possible, impossible or certain: (a) two heads come up; (b) there are three tails; (c) no side comes out; (d) only heads and tails come out. Is there an event more likely than the others are? Which one?" (Edebé. Didactic Guide, 4th grade).

Second, from the items in block 1 (processes, methods, and attitudes in mathematics), the different steps that constitute the problem-solving process were codified. To code this variable, only the items referring to additive structure word problems were taken into account, in whose statement the student was explicitly asked to determine a step corresponding to the solving process, because our interest was focused on verifying to what extent the publishers consider the PS process as an evaluable process. However, the problems that asked the student directly for the result, that is, those problems whose objective was not to assess the different steps of the PS process, were not coded in this content block. These problems were coded in the blocks corresponding to numbering, measurements, geometry, and statistics and probability. Examples:

- Block of numbers (arithmetic). "Solve: Daniela bought for her birthday 8 bags with 15 balloons each one. For the party they inflated 48 balloons. How many balloons did she have left? "(Santillana, Didactic Guide, 4th year).
- Block of measurement (temperature measurements): "Solve: In the town of Luis the temperature was −9 °C at 5 am. Until 11 am, it went up 12 degrees; from 11 am to 4 pm it dropped 6 degrees; and from 4 pm to 10 pm it went up 3 degrees. What was the temperature at 11 am, 4 pm, and 10 pm? (Santillana, Didactic Guide, 6th year).
- Block of geometry: "Solve: Lucía has bought 12 cm diameter circular coasters, and the glasses she has at home are cylinders with a base radius of 0.5 dm. Will the coasters fit her glasses? (H.M, Didactic Guide, 6th year?)
- Block of statistics and probability: "Solve: You throw two coins in the air. Write possible, impossible or certain: (a) two faces come out; (b) there are three crosses; (c) no face comes out; (d) only heads and tails come out. Is there an event more likely than the others are? Which one? (Edebé, Teaching Guide, 4th year).

There were also problems in which it was requested to determine several steps (see Table 1 for the problem corresponding to the example in step 6). In these cases, the problems were coded in as many steps as required in the statement. In the example cited, the problem was coded in the category corresponding to operations (step 4) and verification (step 6).

The coding system was based on the proposal of [52], who conducted an analysis of the ways of solving the problems proposed by the textbooks of the publishers Santillana, Anaya, and SM based on the resolution model of [38].

According to this proposal, seven categories were established, which correspond to the seven steps that make up the solving process (see Table 1). Of these seven categories, the following were considered typical of the superficial solving mode: data, strategies, choice of operations, expression and verification of the result, and invent. For instance, the problem "Manuel's basketball team has made 37 baskets and Pablo's has made 43 baskets. How many baskets does Pablo's team have more than Manuel's does? Read the statement carefully, identify the question and calculate the solution" (Editorial Edelvives), was coded within the superficial mode with three steps: data (read the statement carefully); strategies (identify the question); and operations (calculate the solution).

On the other hand, all combinations of the previous steps that included the reasoning step were considered genuine. For example, the problem "Solve the following problem and explain how you did it: Sandra and Jorge have hung 45 balloons for their little brother's birthday party. If Sandra has hung 25 balloons, how many balloons has Jorge hung?" (Edelvives publisher), was codified within the genuine mode, because the request consists of explaining the entire solving process, including the reasoning, obviously. A similar example is the following: "Last year there were 92 boys and 83 girls at Atlas School. This year there are 210 students, of which 97 are boys. How many more girls are there this year than last year? Show how you did it" (Item of the TIMSS assessment test, 2007, 4th grade. Cognitive domain: reasoning).

**Table 1.** Steps in the PS process.

| Steps | Categories | Examples |
|---|---|---|
| Step 1 | DATA. To manage problem data: extracting relevant data, omitting irrelevant data, and organizing a statement that appears out of order, to extract or to complete data from the problem statement. | Marcos is making a collection that has 25 stickers. If it has 12 stickers, how many stickers does he need to complete the collection? Read the statement carefully (step 1) and calculate the solution (step 4). |
| Step 2 | REASONING. To understand the situation described in the problem: first, in a qualitative way (characters, actions, intentions, etc.) and second, in a quantitative way, establishing the necessary mathematical relationships with the quantities of the problem. | Solve the problem by following all the steps learned (all steps, except invent). Marina inherited a collection of stamps from her grandfather. She has two boxes full of stamps from around the world. There are 11,575 stamps in one box and 12,350 in the other one. Marina bought 5 albums with a capacity of 5000 stamps each. Will she be able to put the entire collection in them? |
| Step 3 | STRATEGIES. To apply strategies that will help to solve the problem: underlining the question, using a drawing or an outline, determining the number of operations necessary to solve the problem, etc. | Underline the question in the following problem (step 3) and solve it later (step 4). Marcos is making a collection that has 25 stickers. If it has 12 stickers. How many stickers does he need to complete the collection? It is missing ___stickers. |
| Step 4 | OPERATIONS. To select the operation or operations necessary to solve the problem. | Carla has bought 11 pieces of mint gum, 13 pieces of strawberry gum, and 9 pieces melon gum. Estimate how many pieces of gum Carla has bought in total (step 3). Calculate how many pieces of gum she has bought in total (step 4). |
| Step 5 | RESULT. Expression of the result. | Underline the most approximated solution (step 5) and check it (step 6). In 1st grade, there are 12 boys and 13 girls. How many students are there in 1st grade? More than 20. Less than 2. |
| Step 6 | VERIFICATION. To check that the result is coherent from a mathematical and situational point of view. | Solve (step 4) and check the result (step 6). Every year Juan's school stages a play to raise funds. Last year 897 people attended the play and this year the audience will be double. The theater has capacity for 1898 spectators. Will everyone be able to enter this year? |
| Step 7 | INVENT. Posing a new problem totally or partially. | Complete the problem with the question (step 7). In a hotel, they expect to host 560 tourists today. There are already 325 settled since yesterday and 136 have arrived this morning. The rest will arrive in the afternoon. |

Source: Own elaboration based on [52].

*2.3. Reliability of the Analysis by Content Blocks and by Steps of the PS Process*

To ensure the coding process, an inter-judge reliability procedure was carried out. First, one of the authors of the study encoded the 1616 items included in the 78 tests analyzed, classifying each item in the corresponding content box. Second, another author carried out independently the coding of 162 items (10% of the total), randomly selected from the set of items included in the unit of analysis and the degree of concordance between both analyses. Reliability, measured with Cohen's Kappa index, was $k = 0.95$. The qualitative interpretation of this index was almost perfect, according to the range of agreement proposed by [53].

Afterwards, four researchers who are Doctors of Education or Educational Psychology codified, independently, 40 items that were selected randomly. Two of the experts were university professors from the field of pedagogy, and the other two were from the field of educational psychology. They were given concise instructions on how to code the items,

but to avoid influencing their work with the possible appearance of biases, the sources from which the items came or the specific objectives of the study were not specified.

The same procedure as in the previous case was followed to calculate Cohen's Kappa index as indicative of the reliability of the four judges. The results showed an agreement index of $k = 0.90$, also almost perfect according to the indications of [53].

Subsequently, to ensure that the procedure for coding the steps of the PS process also had sufficient guarantees of reliability, the two authors independently analyzed the 22 items assigned to the first block of contents (processes, methods, and attitudes in mathematics), specifying the solving steps required in the statement of each problem. Again, Cohen's Kappa index was calculated to verify that the classification was coincident. The result of the reliability between both codifications was $k = 0.92$, also corresponding to the almost perfect level [53].

## 3. Results

### 3.1. Distribution of Items by Content Blocks

The results of the distribution of the 1616 items analyzed in the five content blocks included in the official curriculum are shown in Table 2.

**Table 2.** Results of the frequency and variability of the items by content blocks.

| Publishers | Block 1 | Block 2 | Block 3 | Block 4 | Block 5 |
|---|---|---|---|---|---|
| Santillana | | 197 (62.5%) | 50 (15.9%) | 60 (19%) | 8 (2.5%) |
| Anaya | | 88 (63.3%) | 24 (17.3%) | 22 (15.8%) | 5 (3.6%) |
| S.M | | 55 (62.5%) | 9 (10.2%) | 22 (25%) | 2 (2.3%) |
| Vicens Vives | | 150 (48.2%) | 53 (17%) | 72 (23.2%) | 36 (11.6%) |
| Edebé | | 86 (50.9%) | 27 (16%) | 39 (23.1%) | 17 (10.1%) |
| Edelvives | 22 (3.8%) | 248 (43.1%) | 105 (18.2%) | 152 (26.4%) | 49 (8.5%) |
| Total | 22 (1.3%) | 842 (52.1%) | 268 (16.6%) | 367 (22.7%) | 117 (7.2%) |

B. 1. Processes, methods, and attitudes in mathematics; B. 2. Numbers; B. 3. Measure; B. 4. Geometry; B. 5. Statistics and probability.

The content block that is given the greatest importance in assessment tests is numbers; 52.1% of the items in the tests analyzed are dedicated to assess this type of content. This block is followed by the contents of geometry (22.7%), measurements (16.6%), and the contents of statistics and probability with much lower frequency (7.2%). The first block of contents, referring to the processes, methods, and attitudes in mathematics (block in which the different steps of the PS process are included), was only assessed by one of the six publishers included in the present study (Edelvives), with 22 items, which represents only 1.3% of the total items analyzed. However, it is necessary to clarify that the rest of the publishers included PS tasks (codified in the blocks of numbers, measurements, geometry, and statistics and probability), but not with the aim of assessing the different steps of the solving process, because they were problematic situations in which the students were asked only for the solving and the result. Specifically, Santillana was the publisher with the highest number of problems included in its assessment tests (33.60%). The publishers S.M (25%), Vicens-Vives (23%), and Edebé (20%) followed Santillana with very close percentages. Finally, the publishers with a lower proportion of problems were Anaya (14.38%) and Edelvives (12.60%).

Regarding the analysis by publishers, there were hardly any differences regarding the distribution of the items. The assessment tests included in the teaching guides of all the publishers analyzed prioritized the number block, followed by the blocks of geometry, measurements, and statistics and probability.

### 3.2. Distribution of Items by Solving Mode

After categorizing all the items included in the 78 tests analyzed, a more exhaustive analysis of the 22 items referring to the PS process only (the items included in content block 1) was carried out.

Twelve of the 22 items (54.5%) could be classified into the genuine PS mode (the reasoning or combination of this step with the others was included); and the remaining 10 items (45.5%) corresponded to the superficial mode of PS (reasoning was not included as a strategy to reach the solution).

From these 22 items, 77 steps corresponding to the PS process were identified. The analysis of the data indicated that only 15.5% of the steps corresponded to the reasoning category, while 84.5% of the steps belonged to the different categories of the superficial mode with the following frequency: operations (20.7%); strategies (19.4%); check (18.1%); result (13%), and data (11.6%). Regarding the combination of steps, in the genuine mode the reasoning was included seven times as the only step and five times combined with different categories.

## 4. Discussion

The aims of this study were to analyze which content blocks are given priority in the assessment tests of the mathematics didactic guides; and to analyze how these guides assess the different steps of the PS process.

The results of the distribution of the items by content blocks have shown that the first block of content, referring to mathematical processes, methods, and attitudes, does not seem to be a priority learning objective in the Spanish publishing scene. This is not a very positive finding, considering that previous research has concluded that students' attitudes toward mathematics [54], which are addressed in block 1, and more specifically toward the PS process [55], are closely related to their mathematical performance. In fact, the authors of [55] suggest teaching the students strategies to develop positive attitudes toward PS as a pathway to decrease their mathematical anxiety and to increase their mathematical achievement.

The rest of the content blocks are contemplated in the assessment tests in a disparate way, in some cases according to what is established in the theoretical framework of the TIMSS [26] assessment tests, but in other cases in a notably different way than the one proposed by these tests.

Thus, the block of numbers is assessed with the 52.1% of the total items of the didactic guides and the measurement block with the 16.6% of the total, which are placed in the line of 50% and 15% established, respectively, in TIMSS [26]. The geometry block is overrepresented, because the 22.7% of the items in the assessment tests cover it, and in TIMMS [26] just 15% of these items are proposed. The statistics and probability block is underrepresented, because it is assessed in 7.2% of the items, while in TIMMS 20% is proposed [26].

Regarding the second objective, which refers to the analysis of the type of PS mode that is proposed in the didactic guides, the most remarkable result is that just one of the six editorials reviewed (Edelvives) included items in the assessment tests that allow teachers to know which PS mode students use: that is, if the student applies a superficial solving strategy that excludes reasoning as a key step, or if on the contrary, the student uses a genuine strategy that involves a deep mathematical and situational understanding of the problem. In addition, the items that this publisher presented to assess the PS process are very scarce; they are not raised in a systematic way throughout all the assessment tests, and the strategies corresponding to the superficial and genuine modes were assessed practically equally.

These items are scarce because, as we have seen, of the totality of items presented by this publisher for the assessment of mathematical competence in primary education (576), only 22 (3.82%) are addressed to the solving process. Likewise, they are unsystematic due to the large number of models that this publisher presents (two, three, and four-step models). Finally, if we take into account all the assessed steps, the most frequent categories are those related to the superficial mode, so the step corresponding to reasoning remains in the background. Therefore, these items are also incomplete.

Furthermore, it is paradoxical that publishers include models for learning PS in their textbooks, although they are mostly superficial and unsystematic, as the research carried

out by [52] with the publishers Santillana, Anaya, and SM shows, and, however, the assessment of the PS process is not considered in the tests of the didactic guides designed by those same publishers. In this way, if we take into account that assessment is a key curricular element in the teaching–learning process, there should be a connection between the achievements to be achieved and their assessment. It means that the PS process should be reflected in assessment because students consider important those aspects of instruction that teachers regularly emphasize and evaluate [56]. Indeed, "the assessment system plays a fundamental role because its purposes and methods exert more influence on how and what students learn than any other element of the learning process" [57] (p. 2).

A previous study with textbooks from three publishers (Santillana, Anaya, and SM) [52] concludes that, in general, the solving model proposed to students is characterized by being superficial and unsystematic. It is superficial because most of the models proposed by these publishers do not propose reasoning as a key step in approaching the PS process, and it is unsystematic due to the large number of different models proposed. According to the authors of the study, these results are consistent with the type of problems included in textbooks. In fact, previous studies carried out in Spain with mathematics textbooks [23,58,59] conclude that these curricular materials present problems that are easy and distant from the students' lives, so that reasoning is hardly necessary to solve them. Probably this is one of the reasons that lead students to believe that memorizing formulas, algorithms, and rules is the only method to solve mathematical problems [60].

Furthermore, this study agrees with the previous ones that concluded that PS should be properly addressed in teacher training to improve teachers' knowledge about it and to make them aware of its importance in the teaching and learning process of mathematics, because teachers are the ones who select the tasks that will be performed by the students and the materials that will be used [61,62].

Although this study is based on the research conducted by [52], its purpose is to go one step further, analyzing the PS process in the didactic guides that complement the textbooks, because the tests that assess the math competence of the students appear in these materials. The findings show two fundamental conclusions: the reviewed assessment tests do not distribute their assessment items according to the framework established by TIMSS [26]; and they do not take into account the need to assess in a reasoned and systematic way the PS process that students carry out.

Based on these conclusions, the usefulness of using pre-prepared tests of this type could be questioned. On the one hand, these tests do not seem to have proven to be effective tools to assess the process that students must follow when they are solving problems in a reasoned way. On the other hand, these tests are instruments that exert a notable influence on the conception of both teachers and students when giving meaning to the PS process [56,57], which must be understood as a reflective process of deliberation, and not as a mere calculation exercise where what matters is to arrive mechanically and directly at the solution.

### 4.1. Educational Implications

We conclude with three educational implications that derive from this study, regarding the role of publishers, teachers, and educational administrations.

First, following [47], we would like to comment on two instructional principles that educational research has highlighted: (1) the principle of prolonged commitment, which explains that students must "work on problem tasks on a regular basis, for an extended period of time" [47] (p. 272) in order to improve their PS skills, and (2) the task variety principle, which states that students will improve as problem solvers "only if they are given opportunities to solve a variety of types of problem tasks" [47] (p. 272). According to these two instructional principles, in our opinion, publishers should include in the design and development of their textbooks problem tasks more frequently and systematically. However, it is not only about increasing the amount of problems (that is, a quantitative change), because a qualitative change is also needed, consisting of increasing the variability

of problem situations, which would imply higher levels of challenge. Likewise, publishers should include specific items in their assessment tests that would allow them to assess the process of PS. From our point of view, it is incoherent that textbooks include specific tasks for learning the PS process, but do not include items to assess this process in the assessment tests of their teaching guides.

Second, with regard to teachers, the teaching activity could run between what we could call a "minimum perspective" to another "maximum ideal perspective". On the one hand, from a minimum perspective, the textbook and the didactic guides that complement it could be used by the teacher in a flexible way: that is, to use it as one more among other possible resources, as an aid or as a support to their teaching work, but not as a prescriptive guide to action. On the other hand, from a maximum perspective, the teacher's activity would focus on the elaboration of his own curricular proposals. Now, this second perspective, which would suppose a real change, contrasts with the educational reality: the long and much consolidated tradition of school textbooks. In our opinion, this change could only take place if teachers were aware of the need for it and if they had the necessary knowledge to carry it out: knowledge of mathematics, but also didactic knowledge of mathematics.

Finally, educational policies, beyond the free provision of textbooks, which, as we have pointed out, naturalize the presence of the textbook in school, should be directed both toward quality teacher training (initial and continuous) as well as to the promotion of innovative curricular projects.

### 4.2. Limitations and Future Lines of Research

In this study, we have analyzed the different items proposed by six publishers to assess the mathematical competence of students, through the different content blocks proposed by the current educational legislation in Spain [24]. This analysis has allowed us to determine the relationship between the publishers' proposal and the content dimensions established in the TIMSS theoretical framework [26]. However, this analysis could be extended by also determining the correspondence between the items presented by the publishers and the three cognitive dimensions of TIMSS (knowledge, application, and reasoning), which imply an order of increasing complexity [26].

Moreover, we consider that it would be interesting to complete the results obtained in this study by analyzing in primary education mathematics classrooms the distribution of hours that teachers dedicate to each of the content blocks, with special attention to the hours they dedicate to teach PS processes. This would allow knowing valuable information about what content teachers prioritize, which ones generate more difficulties among students, and what correlation exists between the hours spent on content blocks and the weight these blocks have in the assessment tests.

Another future line of research could be the analysis not only of the content of mathematical problems in primary education, but also the way in which these problems are displayed. It could cover the study of the elements that accompany the textual statement of the problem, such as the pictures, or even the videos in the case of digital textbooks. In this sense, publishers offer teachers the opportunity to present the contents in a more dynamic and flexible way to the students through digital textbooks.

Finally, given that Edelvives has been the only publisher that has included specific items to assess the PS process in the teaching guide tests, it would be interesting to carry out an additional study on the treatment of PS in textbooks of this publisher. Previous studies on the analysis of word problems in the textbooks of Spain have mainly analyzed the publishers Santillana, Anaya, and S.M. At the moment there are no studies from other publishers that are also widely distributed, such as Edelvives, Vicens-Vives, or Edebé.

**Author Contributions:** Conceptualization, J.T.-I. and R.T.-M.; methodology, J.T.-I. and R.T.-M.; investigation, J.T.-I.; data curation, J.T.-I.; writing—original draft preparation, J.T.-I. and I.L.-P.; writing—review and editing, J.T.-I., R.T.-M., and I.L.-P. All authors have read and agreed to the published version of the manuscript.

**Funding:** This research received no external funding.

**Institutional Review Board Statement:** Not applicable.

**Informed Consent Statement:** Not applicable.

**Data Availability Statement:** The data presented in this study are available on request from the corresponding author. The data are not publicly available due to the publishers that edit the teaching guides that have been analyzed in this article maintain the copyright on its content.

**Conflicts of Interest:** The authors declare no conflict of interest.

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
