# Peer review of "Assessment Tests in the Mathematics Teaching Guides in Spain. Analysis of the Content Blocks and the Treatment of Arithmetic Word Problems"

_education, doi:10.3390/educsci11060305_

Round 1

Reviewer 1 Report

Review of the article entitled “Curriculum contents and problem solving models in the assessment tests of the mathematics teaching guides in Primary Education”

This paper, framed within the study of mathematics textbooks, in particular teacher's guides, addresses a topic of interest in the field of research in mathematics education: the analysis of the contents prioritised in textbooks and, in particular, the study of how problem solving is integrated into the initial assessment proposals set out in the teacher's manuals.

Indeed, particularly in the geographical area in which the study was carried out (Spain) and at the level of education studied (Primary Education), teachers are quite dependent on textbooks provided by different publishers. Moreover (and this is an aspect that is not emphasised in the article), it should be noted that in this country, publications are not subject to any kind of control or supervision by the educational authorities to validate their adequacy in terms of educational programmes. This is therefore a very relevant study that needs to be addressed.

The paper presented is a concise and very clearly written work, and it is appreciated that the authors go straight to the issue they are interested in studying and explain it with such clarity. However, despite this clarity, there are some aspects that raise doubts in my mind when reading this paper. In what follows, I will comment on each of the sections of the paper, pointing out both the strong points and those aspects that should be reviewed in depth. In some cases, as will be seen, there are aspects that should be improved that have to do with the design of the study itself and, therefore, it is not reasonable to expect substantial changes, although it would be advisable, with a view to writing a publishable version of the work, for the authors to make an effort to improve the justification of some of the decisions taken in the research design.

Title and abstract

The title of the paper is, in my opinion, rather vague. In fact, when reading the paper, one realises that it does not really analyse "problem solving models" in general, but rather focuses on "arithmetic Word problem solving processes". I recommend that the authors revise the title.

The abstract is clear, frames the work, states the objectives and summarises the results. In relation to the previous comment on the title, it is worth noting that the abstract does not refer to "problem solving models" but rather to "steps of the mathematical problem solving process". Regarding the objectives of the paper, I refer to the concluding comments of my review.

Introduction

The paper does not include a theoretical framework, although both the research and the tools used are detailed in the introduction (correctly organised in two subsections of this part).

I think it would be important for the authors to emphasise in the introduction, based on references, the importance that teachers (particularly primary school teachers) attach to the textbook in mathematics classes. At the end of the review I include some references that may be useful to complement this.

Section 1.1., which describes the current educational programme in Spain at the time of the study, is clear and complete, although it would be useful if the authors could justify their decision to use the TIMSS framework to analyse the distribution of content. Furthermore, given that this section focuses not only on the curriculum framework but also on the latter, it would be useful for the authors to revise the title of the section.

Section 1.2 devoted to "Problem solving models" is, in my opinion, the most problematic (forgive me for the redundance). Here I have missed basic references in the field of mathematics didactics to the work of Polya and Schoenfeld (among others). The authors begin the section by pointing out that the problem-solving process has been approached from two types of theoretical models, but the differences between them are not detailed, or not sufficiently so. It is striking at this point that the authors do not refer to the phases of problem solving process established in Polya's seminal work (which, I understand, would be in line with general models based on heuristics).  On the other hand, the authors talk about cognitive models that refer to verbal arithmetic problems. I understand that in this work they keep this second model, but, to tell the truth, it is not clear at this point (I had to go to reference [12] to understand this part better and, by the way, I detect a degree of similarity that is perhaps excessive).

On the other hand, the differentiation between "superficial" and "genuine" modes of resolution, although I understand that it is a theoretical construct that is well established in the scientific community, it raises many doubts in my opinion. How can we know, solely on the basis of a problem statement, whether or not it fits into the superficial mode? I understand the differences between a superficial resolution and a genuine one, but this will depend on the solver who tackles the problem, so classifying problems based on this categorisation only from the statement seems to me, at least, risky. Anyone who has solved mathematical problems and who has observed students solving problems has seen that, depending on the knowledge and skills of the problem solver, a problem can be anything from a routine activity (in fact, it is no longer a "problem") to a cognitively demanding activity. In this paper, to illustrate the differences between a " superficial" and a " genuine" problem, the authors present two examples, and my question is: is the second problem (in which the key word is, somehow, a "trap") a difficult problem for every person? The answer is, obviously, no, it will only be so for the solver who limits himself to solving by direct translation from natural to arithmetic language word by word, without making sense of the whole statement. Therefore, why classify, only on the basis of the statement, and not even taking into account the profile of the possible solver, this problem as one that requires a "genuine solving mode"? Furthermore, if I think of a first year primary school student who is not yet fully familiar with subtraction, would it not be a challenge for that student to solve the "easy" problem, in which the key word leads to subtraction? I think that, if one wants to categorise these problems, it is better to use a categorisation that does not lead to ambiguities and in which one can classify the problems only from the statement. There are many in the literature (if we want to analyse verbal problems), and, in particular, in relation to the presence or absence of key words in the sentence, we should refer to the work of Schoenfeld (1991).

That said, I believe it would be advisable that the authors revise in depth this section in order to justify and explain more rigorously the theoretical framework adopted in this study.

Methodology

Regarding the methodology section, I have several doubts in the section devoted to describing the analysis methodology. Firstly, in relation to the classification according to the blocks of content, given that block 1 (processes) is transversal to the rest, I find it difficult to understand that the categorisation is univocal (in fact, the article says "For this reason, this block has been formulated with the aim of being part of the daily classroom tasks in order to work on the rest of the contents"). I have the feeling that they have categorised in this block only those activities in which something related to the resolution process is explicitly stated, but of course, any mathematics problem that is posed (and that is really a "problem" for the solvers it is addressed to) implies that the contents of this block 1 are worked on, but also of the others (in the case of the example, I also believe that it falls into block 2). Therefore, if the criterion is really to include in "block 1" only those activities that explicitly mention phases of the problem-solving process, this should be made clear from the start, since it is something central to the study. In fact, I think this should be reviewed, as it is very important for the conclusions drawn from the results.

The following is an explanation of how the problems categorised in Block 1 have been classified. This classification has been used previously in other studies, although, like the theoretical framework of the resolution models, it raises some doubts in my mind. I agree with the authors that what appears in Table 1 are "Step in th PS process" and therefore, as steps, there is a certain relationship between them (you cannot get to some without going through -at least some of-the previous). However, I believe that the description of each category presented in the table is insufficient, perhaps it would be better if the authors developed and justified the choice of this classification in section 1.2 of the introduction (or in a new "theoretical framework" section). For instance, in step 2 they define the process of "reasoning" as "reasoning about the process carried out". This is not the case; the second phase of problem solving involves reasoning about what is to be done, not about what has been done (this would correspond to the last phase, what Polya calls retrospection). On the other hand, it is appreciated that the authors try to illustrate each category with an example, however, there is one example that raises doubts in my mind regarding step 6: in relation to step 6, the authors copy the statement of an activity that says "Solve (step 4) and check the result (step 6). Every year Juan's school stages a play to raise funds. Last year 897 people attended the play and this year the audience will be double. The theatre has capacity for 1898 spectators. Will everyone be able to enter this year?". I wonder whether it is categorised in the "step 4" category because of the word "solve" but, don't you have to read, reason and look for strategies before doing operations?

Finally, at the end of this paragraph it is stated "all combinations of the previous steps that included the reasoning step were considered genuine", I wonder why the academic level of the students is not considered in the categorisation (since their resolutions are not available, at least the relative difficulty of the activity should be considered).  In this sense, in that paragraph the authors refer to the TIMSS categorisation based on cognitive domains, which is interesting, because the TIMSS categorisation allows to classify an activity according to the activity expected by the solver (according to his/her age), while this is not the case in the categorisation proposed in this study.

The part devoted to the reliability of the analysis is clear, and the authors' presentation of these details is appreciated.

In relation to the results, there are, especially in the second part (section 3.2), some things that I find doubtful, but I will not dwell on them here because they are related to the doubts regarding the analysis instruments used. It would be interesting, however, if the authors could provide some examples and more information about those items that have been categorised in block 1 but do not include reasoning in their resolution. This section deserves a major revision.

Discussion

Regarding the discussion of the results concerning the first research objective: the findings are, in my opinion, extremely drastic, and include such strong sentences as "the six publishers analysed systematically ignore the first content block, which refers to the processes, methods and attitudes in mathematics". If this is true, it is extremely important, since it concerns a transversal block of contents. Hence my doubts (which I have already pointed out above), is it possible that there may be activities that have been categorised in a specific block of contents but that intersect with block 1?

Concerning the second objective, I find some confusion in the discussion of results. As formulated in the abstract and at the end of the introduction, the second objective is to identify to what extent these tests evaluate the steps of the mathematical problem solving process. However, in the discussion section, the authors state that the second objective refers to "the analysis of the type of PS model that is proposed in the didactic guides". Indeed, the following lines the authors finally explain clearly what has been done in this study: to analyse to which extent the questionnaires proposed in the teacher's guides include activities that explicitly encourage students to carry out certain procedures linked to the problem solving process (I would add, moreover, that the analysis focuses on verbal arithmetic problems). I recommend the authors to revise, in the light of this comment, both the title and the formulation of the second objective, because (although the objectives formulated in the paper are relevant and interesting) what they actually do is this. This will avoid confusion in the reading of the introduction and methodology sections.

Finally, in relation to the discussion section, I think it would be very interesting for the authors to contrast their results with other studies focusing on the performance and/or attitudes of the students in relation to PS and, even, focusing on the teacher's knowledge of PS, in order to draw conclusions about the implications of the approach of the teacher's guides in the teaching and learning process of mathematics.

The section devoted to the limitations of the study and future lines of research is gratefully acknowledged.

References

González, E. M. L., Guerrero, A. C., & Yáñez, J. C. (2015). La resolución de problemas en los libros de texto: un instrumento para su análisis. Avances de investigación en educación matemática, (8).

Lepik, M., Grevholm, B. & Viholainen, A. (2015). Using textbooks in the mathematics classroom – the teachers’ view. Nordic Studies in Mathematics Education, 20 (3-4), 129-156.

Polya, G. (2004). How to solve it: A new aspect of mathematical method (Vol. 85). Princeton university press.

Schoenfeld, A. H. (1991). On mathematics as sense-making: An informal attack on the unfortunate divorce of formal and informal mathematics. In J. F. Voss, D. N. Perkins, & J. W. Segal (Eds.), Informal reasoning and education (pp. 311–343). Hillsdale, NJ: Lawrence Erlbaum Associates

Schoenfeld, A. H., & Sloane, A. H. (Eds.). (2016). Mathematical thinking and problem solving. Routledge.

Villella, J. A., & Contreras, L. C. (2005). La selección y uso de libros de texto: un desafío para el profesional de la enseñanza de la matemática. Gaceta de la RSME 8(2), 419-433.

Reviewer 2 Report

Lines 19-23: Is a radical orthodoxy and an overtly political statement. The inference that a textbook enforces social or political norms does not necessarily follow as a thesis, and has plenty of critics. Likewise the implication that publishers of books control curricula could be regarded as an almost niche, bordering on conspiracy theory idea. I would strongly suggest rewording this paragraph to tone down.

Line 24-25: This needs supporting citations. It is a very bold claim.

Lines 40-43: The writing here has a post-modern style that is incredibly difficult to understand. Likewise terms such as “macro-projects”, “broadening the gaze” need to be fully explained, for the benefit of the reader. This needs to be changed/reworded/altered for clarity, and possibly expanded in places.

Lines 51-54: Very clear description of the purpose of the manuscript

Lines 55-96: Nice background, well written. Very clear.

Lines 164-247: Methods seem sound to a point and are well described. Have a question regarding the selection of the judges. How does one avoid cognitive biases that may be shared across the set? Were they selected from a wide pool, or a subset available within a single department? Isn’t there a issue associated with potential echo-chamber like effects.

Line 230 & 236: “Next, other author independently” and “In a second moment, four doctor of education” are strangely worded phrases. Suggest changing.

Lines 228-247: Cohen’s kappa is used as a model of concordance between raters. This is only half the story with inter-rater reliability. What was the sensitivity/specificity of the model? Suggest tabulating these results rather than prose.

Lines 249-254: No doubt that there is a big focus on numbers. What is the error/uncertainty in the percentages? And did you consider a more robust form of statistical treatment of the contingency table here?

The conclusions on the implementation of reasoning processes, and lip service paid in tests is shocking.

Lines 357-360: This is the real meat here. I would suggest that to have something of note internationally that you need to conduct this analysis, as it would provide a generalisable framework for interpreting the interactions between educators, publishers and subject guidelines.

Reviewer 3 Report

80. Consider using the plural form of "polyhedron" (either polyhedra or polyhedrons).

This is a very well presented research study and will no doubt be of great interest to the stakeholders within primary mathematics education in Spain.

A brief discussion as to the emergence of online resources would have been welcome as would the publishers contribution to the more dynamic (and thus easily amended) online medium.

Briefly considering that the way in which content is delivered may have as much bearing as the assessment content itself would add some depth of understanding to the landscape the teaching and learning of mathematics in a primary setting.

Reviewer 4 Report

The paper is devoted to the type of tasks used in Spanish mathematics teaching guides and how proposed tasks in assessment tests are in the coherence with tasks in the international study TIMSS. It is also analysed how these special methodical materials for primary math teachers in Spain support the steps of the mathematical problem-solving process.

I have some recommendations:

  1. In the part 1.2 on the page 3 is possible to add some information about steps in problem solving special for mathematics education (see Special Issue https://www.mdpi.com/journal/sustainability/special_issues/math_education_problem-solving). I think the theory of Polya or Schoenfeld in the problem solving is important (see paper https://www.mdpi.com/2071-1050/12/23/10113)
  2. The table 2 on the page 5 is difficult to read, it will be better to add vertical and horizontal lines for each row and column.
  3. I recommend adding the information on the page 2 line 90, why authors have the statement, that it is important implement some results or ideas from study TIMSS to methodical materials for Spanish teachers and not something from another study. Why is TIMSS important? It is possible to add also short information abut another study, if except TIMSS exists for paper purposes.
  4. It is written on the page 7 in discussion that some blocks are underrepresented, and some blocks are overrepresented. Is this fact in coherence with number of teaching hours at Spanish primary schools, that overrepresented blocks obtain at Spanish primary schools more teaching hours? On the page 7 line 295 is written “The geometry block is overrepresented... “. Is it something wrong? Many studies in some European countries state, that there exist not enough geometry teaching hours or geometry tasks at schools.
  5. I recommend in further research (page 9, part 4.1) to think about number of teaching hours at schools for mentioned blocks in the paper. I think this discussion is also about priorities in the educational process. It is important to give attention, that some mathematical notions are more difficult, complicated, or more complex and they need more time in mathematics education. Has this kind of research the ambition to formulate some educational recommendations for primary math teachers in their practise?

Round 2

Reviewer 1 Report

I appreciate the effort made by the authors for revising their work following the review.

The work has been substantially improved, in particular with the revision of some key aspects such as the formulation of the second research objective. 
The sections of the introduction that constitute the theoretical framework of the paper have been considerably improved, particularly the one devoted to problem-solving models. The difference between "superficial" and "genuine" is now clear, as it does not refer to the problem itself but to the resolution mode. However, it is important that the authors check the use of the words "mode" and "model" throughout the work, as there are still some typos (see, for example, lines 365 and 361).

The section on the methodology of analysis is also much clearer now, the categories used in the classification are straightforward and well defined.

A laudable effort has been made in revising the text of the discussion section, which has also been enriched by the introduction of a section dedicated to the educational implications of the findings.

I would like to congratulate the authors for their work and thank the editors for giving me the opportunity to review this paper.

Author Response

Dear Reviewer,

We really thank you for appreciating our effort to improve the quality of the article based on your first suggestions. Regarding the following suggestion:

“The difference between "superficial" and "genuine" is now clear, as it does not refer to the problem itself but to the resolution mode. However, it is important that the authors check the use of the words "mode" and "model" throughout the work, as there are still some typos (see, for example, lines 365 and 361)”.

We have changed the word “model” by “mode” in lines 356, 361, 365 and 480. Moreover, we have reviewed the use of these words in the rest of the article and we consider that is all right now.

Thanks again.

Best regards.

Reviewer 2 Report

Authors have addressed the questions I put forward reasonably. I would suggest that the paper needs to be sense checked for syntax/grammar and whatnot. Additionally the references are quite messy at the moment, and need fixing.

Author Response

Dear Reviewer,

We really thank you for appreciating our effort to improve the quality of the article based on your first suggestions. Below we expose the changes made in the article text (they are also marked on the document using the “Track Changes” function) according to the new suggestions.

“I would suggest that the paper needs to be sense checked for syntax/grammar and whatnot”. 

We agree with this suggestion, so we have checked the syntax, grammar, and vocabulary throughout the article and we have made some minor changes that we think may improve the text.

“Additionally the references are quite messy at the moment, and need fixing”.

According to the instructions of the journal, we have fixed all the references that were not fully correct.

Best regards.